# Convergence of pontine and proprioceptive streams onto multimodal cerebellar granule cells

Cheng-Chiu Huang[1], Ken Sugino[1], Yasuyuki Shima[2], Caiying Guo[1], Suxia Bai[1], Brett D Mensh[1], Sacha B Nelson[2], Adam W Hantman[1]*

[1]Janelia Farm Research Campus, Howard Hughes Medical Institute, Ashburn, United States; [2]Department of Biology and Center for Behavioral Genomics, Brandeis University, Waltham, United States

**Abstract** Cerebellar granule cells constitute the majority of neurons in the brain and are the primary conveyors of sensory and motor-related mossy fiber information to Purkinje cells. The functional capability of the cerebellum hinges on whether individual granule cells receive mossy fiber inputs from multiple precerebellar nuclei or are instead unimodal; this distinction is unresolved. Using cell-type-specific projection mapping with synaptic resolution, we observed the convergence of separate sensory (upper body proprioceptive) and basilar pontine pathways onto individual granule cells and mapped this convergence across cerebellar cortex. These findings inform the long-standing debate about the multimodality of mammalian granule cells and substantiate their associative capacity predicted in the Marr-Albus theory of cerebellar function. We also provide evidence that the convergent basilar pontine pathways carry corollary discharges from upper body motor cortical areas. Such merging of related corollary and sensory streams is a critical component of circuit models of predictive motor control.

## Introduction

The cerebellum is critical for coordinating movement and for learning sensorimotor relationships (*Ito, 2006*, *2008*). Sensory and motor-related afference to the cerebellum is largely conveyed by mossy fiber inputs to granule cells, which notably constitute over half the neurons of the mammalian brain (*De Schutter and Bjaalie, 2001*; *Herculano-Houzel and Lent, 2005*; *Watson et al., 2012*). Granule cells distribute this sensory and motor information to the rest of cerebellar cortex through crystalline circuitry which has been well-characterized over the past 100 years (*Sotelo, 2008*, *2011*). However, the nature of the computations which granule cells perform on incoming afferents remains unresolved.

The input structure to granule cells constrains their potential functions. Granule cells are simple neurons, having on average only four dendrites, each of which receives a single, large, excitatory mossy fiber input (*Gray, 1961*; *Eccles et al., 1967*; *Palay and Chan-Palay, 1974*; *Jakab and Hamori, 1988*). Mossy fibers project widely in the cerebellar cortex from a disparate set of sensory and motor-related relay structures throughout the brainstem and spinal cord. Knowing whether multiple types of mossy fibers synapse onto the same granule cell is key to understanding the types of operations they perform (*Ekerot and Jorntell, 2008*; *Arenz et al., 2009*). For example, a granule cell receiving all synaptic inputs from the same mossy fiber source (i.e., 'unimodal') may serve to filter noise by requiring the summation of inputs in order to fire action potentials (*Jorntell and Ekerot, 2006*; *Ekerot and Jorntell, 2008*; *Bengtsson and Jorntell, 2009*). A multimodal arrangement, in which an individual granule cell mixes inputs from different mossy fiber origins (e.g., one sensory and one

*For correspondence:
hantmana@janelia.hhmi.org

**eLife digest** Learning a new motor skill, from riding a bicycle to eating with chopsticks, involves the cerebellum—a structure located at the base of the brain underneath the cerebral hemispheres. Although its name translates as 'little brain' in Latin, the cerebellum contains more neurons than all other regions of the mammalian brain combined.

Most cerebellar neurons are granule cells which, although numerous, are simple neurons with an average of only four excitatory inputs, from axons called mossy fibers. These inputs are diverse in nature, originating from virtually every sensory system and from command centers at multiple levels of the motor hierarchy. However, it is unclear whether individual granule cells receive inputs from only a single sensory source or can instead mix modalities. This distinction has important implications for the functional capabilities of the cerebellum.

Now, Huang et al. have addressed this question by mapping, at extremely high resolution, the projections of two pathways onto individual granule cells—one carrying sensory feedback from the upper body (the proprioceptive stream), and another carrying motor-related information (the pontine stream). Using a combination of genetic and viral techniques to label the pathways, Huang and co-workers identified regions where the two types of fiber terminated in close proximity. They then showed that around 40% of proprioceptive granule cells formed junctions, or synapses, with two (or more) fibers carrying different types of input. These cells were not uniformly distributed throughout the cerebellum but tended to occur in 'hotspots'.

Lastly, Huang et al. examined the type of information conveyed by the sensory and motor-related input streams whenever they contacted a single granule cell. They confirmed that when the sensory input consisted of feedback from the upper body, the motor input consisted of copies of motor commands related to the same body region. Because it is thought that the cerebellum converts sensory information into representations of the body's movements, directing motor commands to these same circuits may allow the cerebellum to predict the consequences of a planned movement prior to, or without, the actual movement occurring.

The work of Huang et al. provides evidence to support the previously controversial idea that granule cells in the mammalian cerebellum receive both sensory and motor-related inputs. The labeling technique that they used could also be deployed to study the inputs to the cerebellum in greater detail, which should yield new insights into the functioning of this part of the brain.

motor-related precerebellar nucleus), enables more complex functions. Such mixing of mossy fiber inputs was a critical aspect of the expansion recoding of cerebellar afference performed by granule cells in the influential Marr-Albus theory of cerebellar function (*Marr, 1969*; *Blomfield and Marr, 1970*; *Albus, 1971*). If granule cells can indeed be multimodal, mapping these convergences across the cerebellum will be critical in uncovering their role in particular cerebellar functions.

Controversy surrounds the multimodal capacity of granule cells (*Ekerot and Jorntell, 2008*; *Arenz et al., 2009*). Electrophysiological evidence was found for multimodal granule cells in the electrosensory lobe of the mormyrid fish (*Sawtell, 2010*). Whether this granule cell feature extends to the cerebellum and to the mammal remained to be determined. Recently, mammalian granule cell input structure has been tested by in vivo receptive field mapping studies (*Jorntell and Ekerot, 2006*; *Arenz et al., 2008*; *Bengtsson and Jorntell, 2009*). Such studies in the mammal have failed to positively identify more than one input source converging onto individual cerebellar granule cells (*Jorntell and Ekerot, 2006*; *Arenz et al., 2008*; *Bengtsson and Jorntell, 2009*). This lack of evidence for multimodality has led to a diminution of models of granule cell function from a recoder to a noise filter (*Ekerot and Jorntell, 2008*). However, these electrophysiological studies did not systematically examine granule cells across cerebellar cortex, nor did they test the most numerous mossy fiber inputs, those originating from the basilar pontine nucleus (BPN) (*Brodal and Bjaalie, 1992*).

The majority of BPN input originates from output messages of the cerebral cortex, superior colliculus, red nucleus, and other motor centers (*Burne et al., 1981*; *Mihailoff et al., 1989*; *Panto et al., 1995*; *Schwarz and Thier, 1999*; *Leergaard et al., 2000*; *Leergaard, 2003*; *Tziridis et al., 2012*). Therefore BPN mossy fibers are in a position to carry copies of motor commands—corollary discharges—into the

cerebellum (*Sperry, 1950*; *von Holst and Mittelstaedt, 1950*; *Poulet and Hedwig, 2007*; *Glickstein and Doron, 2008*; *Sommer and Wurtz, 2008*). In the central nervous system, corollary discharges are found in circuits involving the intersection of motor and sensory pathways (*Wolpert and Miall, 1996*; *Sommer and Wurtz, 2008*). While the granular layer receives both of these types of mossy fiber inputs, the controversy about the multimodal nature of individual granule cells raises questions about their capacity to mediate the intersection of pontine and sensory pathways at the cellular level.

To test both the multimodal nature of granule cells and their specific role in merging pontine and sensory streams, we examined the intersection of the BPN pathway and a primary sensory precerebellar pathway related to motor output—forelimb and upper-trunk proprioceptive information projecting to the cerebellum through the external cuneate nucleus (ECN) of the hindbrain (*Campbell et al., 1974*; *Akintunde and Eisenman, 1994*; *Quy et al., 2011*). Combining genetics, viral tracing, and large scale confocal microscopy allowed us to take advantage of the unique mossy fiber/granule cell structure to generate synapse-resolution maps of ECN and BPN projections across the entire cerebellar cortex. We found that ECN and BPN inputs synapse onto the same granule cells, with a reproducible, region-specific, cerebellar topography. For a cerebellar area receiving upper body proprioceptive information, we show that the BPN input receives cortical afferents from an area associated with upper body motor control. Therefore, pontine and proprioceptive streams related to somatotopically similar motor output commands may integrate in multimodal granule cells of the cerebellum.

## Results

### Co-termination of sensory and pontine inputs in the cerebellum

To explore the intersection of sensory and pontine pathways, we used a combined genetic/viral strategy to trace the projection patterns of ECN and BPN inputs to the cerebellum. The genetic component of the strategy was used to distinguish the ECN and BPN from other nearby precerebellar sources, and the viral component was used to distinguish them from each other. First we searched for genes exhibiting regional selectivity for both the ECN and the BPN. Literature and Allen Institute Anatomic Gene Expression Atlas database searches produced candidates fitting this expression profile (*Hisano et al., 2002*; *Ng et al., 2009*, *2010*). The *solute carrier family 17 (sodium-dependent inorganic phosphate cotransporter), member 7 (Slc17a7)* gene was selected and a knock-in mouse, *Slc17a7-IRES-Cre*, expressing Cre under the control of this locus was generated. The Cre-dependent reporter expression in this mouse recapitulates the selective expression pattern of the *Slc17a7* locus as reported in the Allen Institute Anatomic Gene Expression Atlas database (*Figure 1—figure supplement 1*). Taking advantage of selective Cre expression and the approximate 4-mm separation between the ECN and the BPN, we stereotaxically injected different Cre-dependent reporter viruses into each nucleus of the *Slc17a7-IRES-Cre* mice (*Figure 1A*). This strategy resulted in selective and distinguishable labeling of the ECN and the BPN (*Figure 1B*). Axons of ECN and BPN were intensely labeled (*Figure 1B*, white arrowheads) and could be identified at their terminations in the cerebellum (*Figure 1C,D*).

Distinguishable labeling of the ECN and the BPN allowed examination of the spatial relationship of these cerebellar afferent systems. As has been previously reported, at a gross level the ECN and BPN target mostly non-overlapping regions of the cerebellum (*Akintunde and Eisenman, 1994*; *Serapide et al., 1994*, *2001*). ECN inputs primarily project to the ipsilateral cerebellar vermis and BPN inputs bilaterally terminate in the cerebellar hemispheres (*Figure 1C,D*). However, strong viral-reporter expression and high-resolution, large-area confocal microscopy revealed many areas of the cerebellum where ECN and BPN axons co-terminate. Such areas include the vermis, lateral vermis, simple lobule, paraflocculus, Crus1, Crus2, paramedian lobule, and copula of the pyramis (*Figure 1D*). Within these areas, ECN and BPN axons terminate within very close proximity (<50 μm), well within the length of the dendritic arbors of granule cells (*Figure 1D*, inset). Therefore, various cerebellar microcircuits potentially process both proprioceptive and pontine information.

### Cellular locus of ECN and BPN convergence

Co-termination within the spatial range of granule cell dendrites does not necessarily indicate the convergence of ECN and BPN inputs onto the same granule cell. A mossy fiber terminal synapses with granule cell dendrites within a glomerulus (*Ekerot and Jorntell, 2008*; *Arenz et al., 2009*). Inside a glomerulus, the claw-footed dendrites of granule cells wrap around and make multiple synaptic contacts

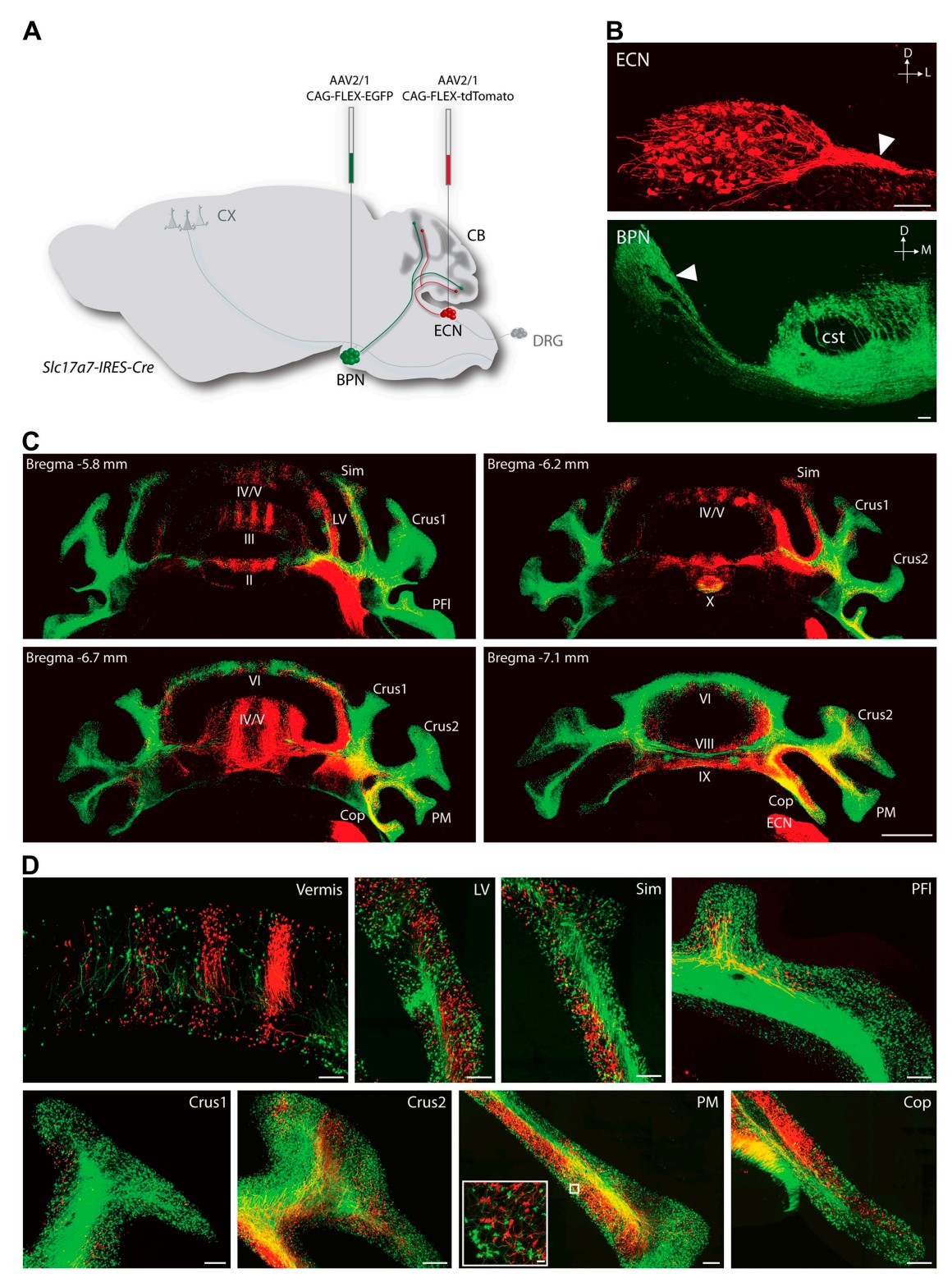

**Figure 1**. Termination patterns of ECN and BPN mossy fibers in the cerebellum. (**A**) Genetic and viral scheme to specifically label ECN and BPN mossy fibers. Cre-dependent AAVs expressing *tdTomato* and *EGFP* are stereotaxically injected into ECN and BPN respectively in the *Slc17a7-IRES-Cre* mouse brain. CX, cortex; CB, cerebellum; DRG, dorsal root ganglia. (**B**) Confocal images of viral injection sites (ECN and BPN). White arrowheads, ECN and BPN axonal tracts; cst, corticospinal tract. D: dorsal; L: lateral; M: medial. Scale bars, 100 µm. (**C**) Projection pattern of ECN (red) and BPN (green) mossy fibers
*Figure 1. Continued on next page*

*Figure 1. Continued*

in the cerebellum. Montage confocal images of the cerebellum from rostral to caudal (Bregma −5.8 to −7.1 mm) positions. Vermis (II, III, IV/V, VI VIII, IX, X); Copula of the pyramis (Cop); lateral vermis (LV); Paraflocculus (PFl); Paramedian lobule (PM); simple lobule (Sim). Scale bar, 1 mm. (**D**) Magnified co-termination fields of ECN (red) and BPN (green) mossy fibers in selected cerebellar lobules. Boxed area shows high density of ECN and BPN mossy fiber terminations in the paramedian lobule. Scale bars, 100 µm; 10 µm in boxed area.
The following figure supplements are available for figure 1:

**Figure supplement 1**. Expression pattern of *Slc17a7-IRES-Cre* mouse line.

with the single, large, mossy fiber termination. This synaptic arrangement allows us to determine, with light microscopy, whether ECN and BPN mossy fibers converge onto single granule cells. Anatomical assessment of convergence requires simultaneous and separate labeling of granule cell morphology and of the two types of mossy fibers. The genetic/viral approach described in *Figure 1A* accomplishes the labeling of the ECN and BPN mossy fibers. However, cell density and the spatial expanse of the cerebellum precluded a similar approach for revealing the morphology of individual granule cells throughout the cerebellum. Alternatively, we screened enhancer-trap mouse lines for an expression pattern where granule cells are labeled in a sparse, 'Golgi stain-like' fashion. This screen identified one mouse line, *TCGO*, which expresses *tetracycline transactivator* and *mCitrine* in a subset of granule cells. While the density of labeled cells is consistent across *TCGO* animals, we lack evidence to distinguish if the labeling is stochastic or marks a particular class of granule cells. Strong expression of the *mCitrine* reporter, a low density of marked cells, and labeling spanning all cerebellar cortical regions allowed manual segmentation of the dendritic morphology of neighboring granule cells over the entire expanse of the cerebellum (*Figure 2A–C*). Simultaneous labeling of the two types of mossy fibers and the granule cells was accomplished by injecting Cre-dependent viruses into both the ipsilateral ECN and the contralateral BPN of *Slc17a7-IRES-Cre; TCGO* bitransgenic animals (*Figure 2D*).

To test the convergence of sensory and pontine pathways, we first identified proprioceptive granule cells, defined as those with at least one input originating from the ECN. Once classified, we next tested if these proprioceptive granule cells also synapse with BPN mossy fibers. In areas of ECN and BPN co-termination, the dendritic morphology of each proprioceptive granule cell was traced and the identity of any unambiguous synaptic input was cataloged. Based on results from one annotator, 40% of proprioceptive granule cells (2429/5997 in two animals) made obvious synapses (see 'Materials and methods') with both ECN and BPN inputs (*Figure 2E*). Therefore, subsets of ECN and BPN information streams converge at their first opportunity in the cerebellum—onto granule cells.

## Survey of ECN and BPN convergence across the cerebellum

Because of the known functional specialization of cerebellar subregions (*Chambers and Sprague, 1955a*, *1955b*; *Apps and Garwicz, 2005*; *Cerminara and Apps, 2011*), we surveyed where ECN-BPN convergence occurs, and does not occur, throughout the cerebellum. We were able to generate such maps due to the cerebellum-wide expression of the *TCGO* transgene and the ability to image large expanses of cerebellar neuropil. For each cerebellar region receiving input from the ECN (*Figures 3–5*), two locations along the anterior-posterior axis were analyzed. Each proprioceptive granule cell was classified into one of three classes (*Figure 2F–H*): those receiving one ECN input only (*E*), those receiving at least one ECN and at least one BPN input (*EB*), and those receiving more than one ECN input but no BPN inputs (*E$^+$*). Granule cells without ECN inputs but synapsing with one or multiple BPN axons (*B*, *B$^+$*, respectively) were only analyzed for select sections due to the abundance of such granule cell types. Both mouse brains were examined by two annotators; the pattern and percentage of granule cell types largely agreed across animals and annotators (*Figures 3–5*).

We observed three types of granule cell terrain across cerebellar areas. Some cerebellar areas, such as medial vermis, were composed mostly of *E and E$^+$* granule cell (*Figure 3A,B*). In some cases the lack of EB cells was due to the scarcity of BPN inputs to the region (compare *Figure 3A,C*), but in other regions the number of granule cells receiving BPN inputs (*B*, *B$^+$*) was roughly equivalent to the number receiving ECN inputs (compare *Figure 3B$_{cb1}$*, *Figure 3D$_{cb1}$*). Therefore, vermal areas harbor proprioceptive pathways which are segregated from BPN inputs at the cellular level; this segregation would not be evident with traditional neuro-anatomical techniques. Other areas *intermingle all three granule cell types*, this category includes lateral vermis, simple lobule, copula of the pyramis and Crus1 (*Figure 4*). The densities of

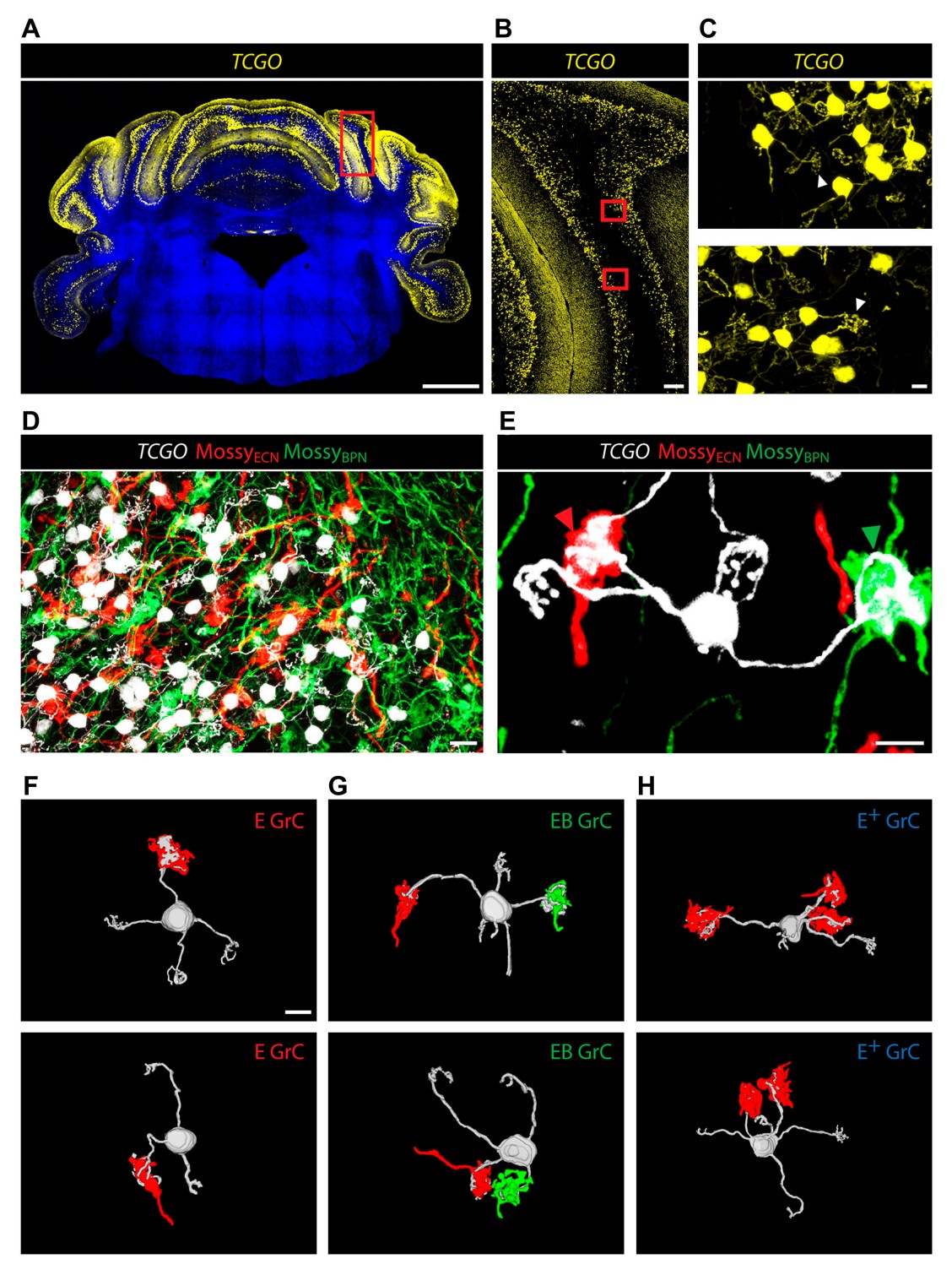

**Figure 2**. Convergence of ECN and BPN mossy fibers on cerebellar granule cells. (**A**) *TCGO* transgene expression in a representative section of cerebellum. Scale bar, 1 mm. (**B**) *TCGO mCitrine* expression in boxed area of (**A**), simple lobule. Scale bar, 100 μm. (**C**) Maximum projection of labeled granule cells in *TCGO* mice (white arrowhead: dendritic arborization) in boxed areas of (**B**). Scale bar, 5 μm. (**D**) Co-termination of ECN (red) and BPN (green) mossy fibers in paramedian lobule of a *Slc17a7-IRES-Cre; TCGO* mouse. Scale bar, 10 μm. (**E**) Maximum projection of a labeled granule cell that receives mossy fiber inputs from ECN (red arrowhead) and BPN (green arrowhead) in a *Slc17a7-IRES-Cre; TCGO* mouse. Scale bar, 5 μm. (**F**)–(**H**) 3D reconstruction of granule cells with associated mossy fiber terminations. *E* granule cell (GrC), granule cell with one ECN input and one other traceable dendrite; *EB* GrC, granule cell with ECN and BPN input(s); *E*⁺ GrC, granule cell with two or more ECN inputs but no BPN input. Scale bar, 5 μm.

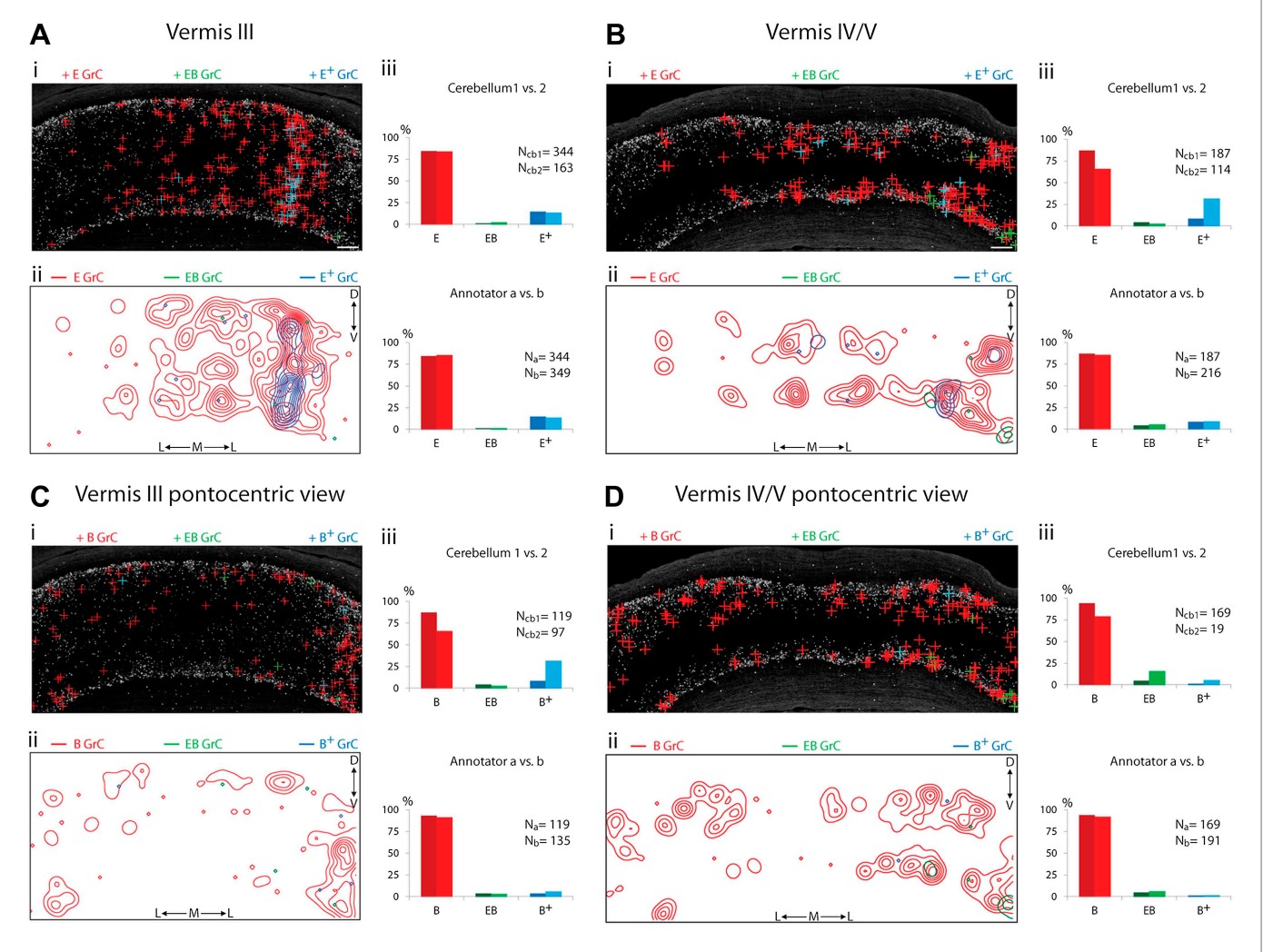

**Figure 3**. Cerebellar areas not exhibiting convergence of sensory and pontine inputs. Survey of ECN and BPN convergence in the anterior vermis. (**A**) Vermis III. (**B**) Vermis IV/V. (i) Granule cell (GrC) classification and distribution. Red cross, *E* GrC; green cross, *EB* GrC; blue cross, *E+* GrC. Scale bars, 100 μm. (ii) density contour map of *E*, *EB* and *E+* granule cells. D: dorsal; V: ventral; M: medial; L: lateral. Red, green and blue lines in the contour map represent density of E, *EB,* and *E+* granule cells respectively. (iii) upper, percentage of *E*, *EB* and *E+* granule cells of two *Slc17a7-IRES-Cre; TCGO* cerebella. Lower, comparison between annotators in percentage of *E*, *EB* and *E+* granule cells in a selected section. (**C**) Pontocentric view of vermis III. (**D**) Pontocentric view of vermis IV/V. (**C** and **D**) Same organization as in (**A** and **B**) but *B* replaces *E* and *B+* replaces *E+* granule cells. *B* GrC: granule cell with one BPN input and one other traceable dendrite; *B+* GrC: granule cell with two or more BPN inputs but no ECN input. *EB* GrC is the same as in (**A** and **B**)

granule cell type varied in these areas along the dorsal-ventral, and medial-lateral axes. Thus, these areas deliver multiple types of information to the overlying Purkinje cells and the combinations of these types can vary locally. Lastly, some cerebellar areas are *dominated by EB granule cells*. These areas include paramedian lobule, the paraflocculus, and Crus2 (*Figure 5*). These areas are 'hotspots' for the intersection of ECN and BPN pathways, where the majority of proprioceptive granule cells also receive BPN inputs. Consequently, many parallel fiber inputs to Purkinje cells in these areas carry both ECN and BPN information. Differences in the extent and patterns of ECN and BPN convergence suggest that the cerebellum handles proprioceptive information with regional specificity.

## Source of cortical inputs to cerebellar proprioceptive pathways

ECN inputs carry forelimb and upper body proprioceptive information to cerebellar granule cells (*Campbell et al., 1974*; *Akintunde and Eisenman, 1994*; *Quy et al., 2011*). We tested if the BPN inputs that converge with ECN signals are capable of delivering corollary discharges relating to similar regions

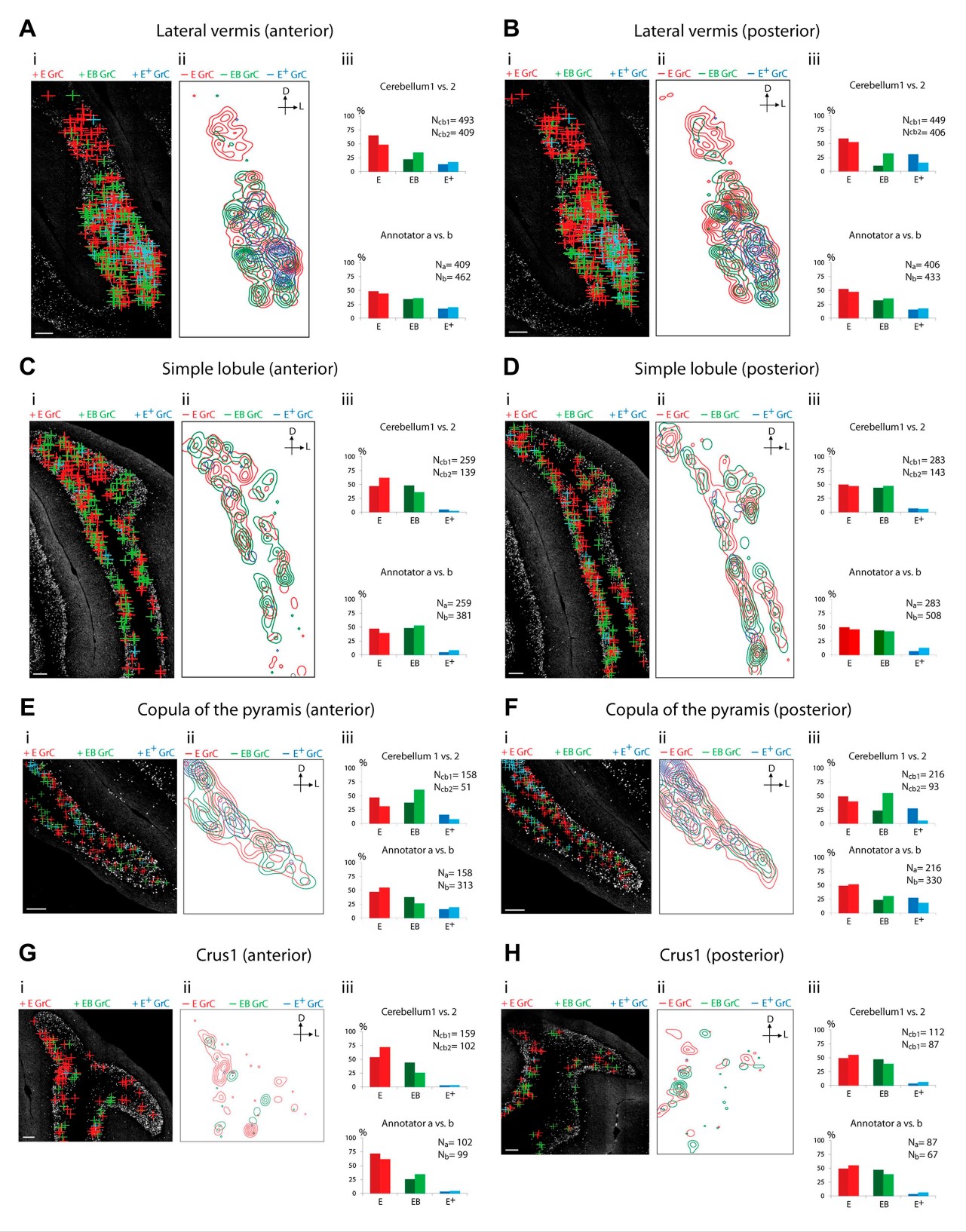

**Figure 4**. Cerebellar areas exhibiting mixtures of convergence and separation of sensory and pontine inputs. Survey of ECN and BPN convergence in the lateral vermis, simple lobule, copula of the pyramis and Crus1. (**A**) Lateral vermis, anterior section. (**B**) Lateral vermis, posterior section. (**C**) Simple

*Figure 4. Continued on next page*

*Figure 4. Continued*

lobule, anterior section. (**D**) Simple lobule, posterior section. (**E**) Copula of the pyramis, anterior section. (**F**) Copula of the pyramis, posterior section. (**G**) Crus1, anterior section. (**H**) Crus1, posterior section. (i) Granule cell (GrC) classification and distribution. Red cross: *E* GrC; green cross: *EB* GrC; blue cross: $E^+$ GrC. Scale bars, 100 μm. (ii) Density contour map of *E*, *EB* and $E^+$ granule cells. D: dorsal; L: lateral. Red, green and blue lines in the contour map represent density of *E*, *EB,* and $E^+$ granule cells, respectively. (iii) Upper, percentage of *E*, *EB* and $E^+$ granule cells of two *Slc17a7-IRES-Cre; TCGO* cerebella. Lower, comparison between annotators in percentage of *E*, *EB* and $E^+$ granule cells.

of the body. To demonstrate the cortical nature of pontine inputs, we combined anterograde labeling from forelimb/upper body primary motor cortex (M1) and retrograde labeling of BPN neurons (*Figure 6A*). We focused on the paramedian lobule as the projection target since nearly every paramedian, proprioceptive granule cell receives BPN inputs. For retrograde labeling, we employed a mCherry-expressing ASLV-A envelope glycoprotein (EnvA) pseudotyped, glycoprotein-deleted rabies virus (SADΔG-mCherry(EnvA)), whose tropism is restricted to *avian tumor virus receptor A* (*TVA*)-expressing cells (*Wickersham et al., 2007*; *Wall et al., 2010*). Cre-dependent viral expression of *TVA* in the BPN in *Slc17a7-IRES-Cre* mice selectively sensitizes BPN neurons to SADΔG-mCherry(EnvA) infection. To selectively label paramedian-projecting BPN neurons, we forced SADΔG-mCherry(EnvA) rabies infection to originate from BPN distal axons by delivering the rabies virus to the paramedian lobule. To anterogradely label M1, we stereotaxically injected Cre-dependent *EGFP*-expressing viruses into a region of M1 known to contain neurons which drive forelimb/upper body movement (*Ayling et al., 2009*; *Harrison et al., 2012*). EGFP-labeled neurons were restricted to lateral agranular cortex, consistent with M1 identity (*Figure 6B*) (*Tennant et al., 2011*). We then examined if motor cortical axons synapse with paramedian-projecting BPN neurons. The majority of pontine inputs to the paramedian lobule originate from the medial-ventral BPN (*Figure 6C*, upper). Retrogradely-labeled BPN neurons are situated in dense fields of M1 axons (*Figure 6C*, lower; *Figure 6D*). Cortical axons containing presynaptic vesicle proteins (*Bellocchio et al., 1998*) were in close proximity to retrogradely labeled pontine neurons; these M1 terminals were also associated with post-synaptic densities of BPN neurons (*Naisbitt et al., 1999*) (*Figure 6E*). Putative M1 synaptic inputs were identified on rabies-labeled BPN neurons (minimum 10 synaptic partners analyzed per mouse) in 3/3 animals. We did not attempt to quantify the percentage of retrogradely labeled neurons receiving M1 inputs due to the high false negative rate originating from the intentionally incomplete labeling of cortical axons and the inefficiencies of immunostaining synaptic proteins. Taken together, these findings suggest synaptic arrangements between forelimb/upper body motor cortex and paramedian-projecting pontine neurons. Therefore, forelimb/upper body pathways separately carrying corollary discharges and proprioceptive information are aligned in cerebellar cortex.

## Discussion

Our results indicate that axonal connections carrying pontine information directly converge with proprioceptive pathways onto individual granule cells. The percentage of proprioceptive granule cells exhibiting such convergence varied across cerebellar areas. In an area exhibiting a high degree of convergence, granule cells receive BPN inputs with the capability of carrying motor cortical corollary discharges. These findings establish the multimodal nature of granule cells and locate a merging of sensory and corollary pathways in specific cerebellar subregions.

### Multimodal nature of granule cells

Since the number of granule cells far exceeds the number of their mossy fiber inputs, granule cells are in a position to potentially permute combinations of afferent inputs in the cerebellum (*Marr, 1969*; *Blomfield and Marr, 1970*; *Albus, 1971*). David Marr and James Albus proposed that by mixing mossy fiber inputs, granule cells could perform such 'expansion recoding', enabling them to contribute to pattern separation and consequently to be fundamental for motor learning (*Marr, 1969*; *Blomfield and Marr, 1970*; *Albus, 1971*). In this model, the associative capacity of granule cells is maximized if different modalities of mossy fibers are mixed onto individual granule cells. To test potential mixing of granule cell inputs, several groups performed receptive field mapping studies. In the C3 region of the anterior paravermis of the cerebellum, failure to find evidence for mixing led Jorntell and Ekerot to disfavor the multimodal view and relegate granule cell function from expansion recoding to noise

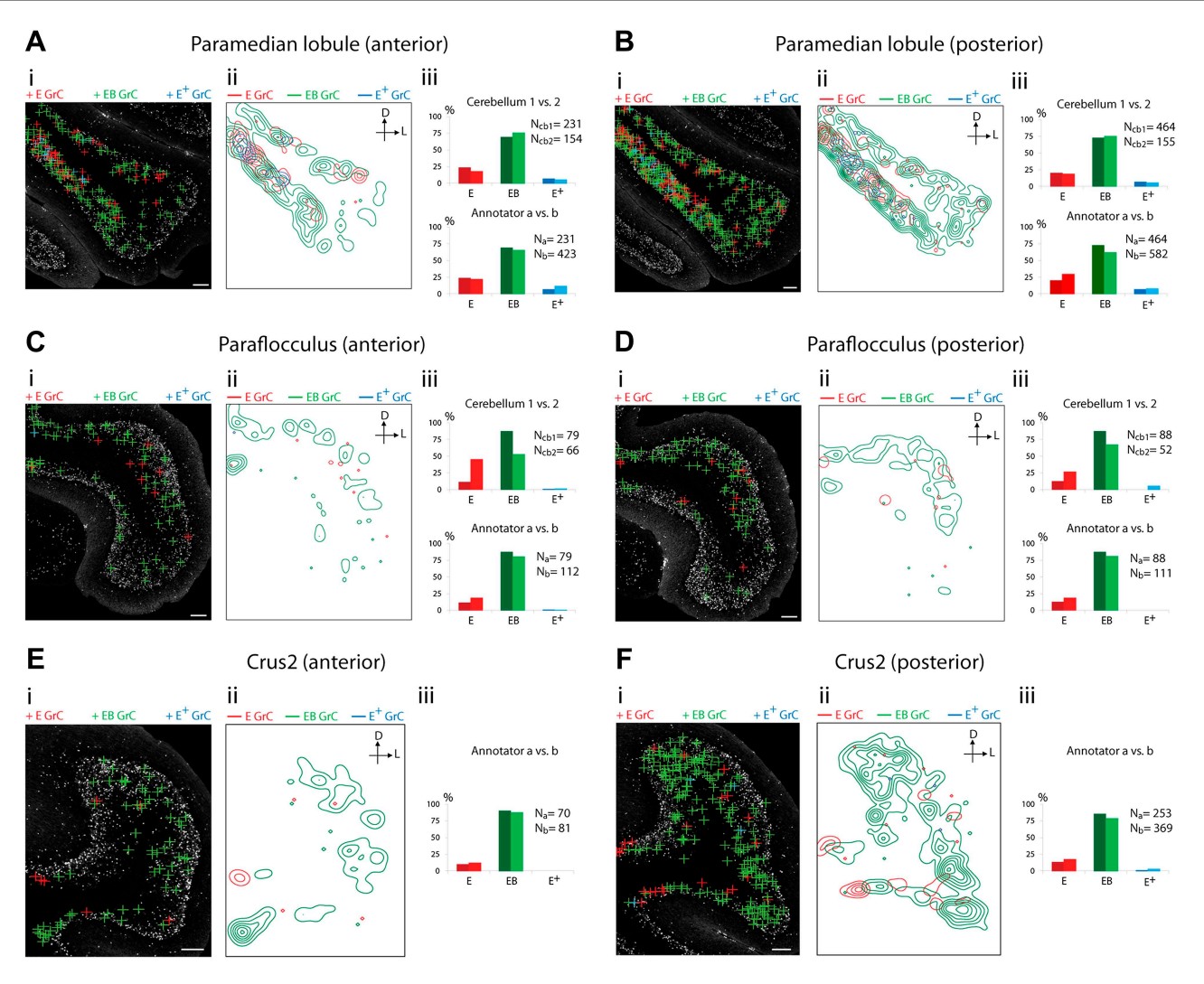

**Figure 5**. Cerebellar areas exhibiting abundant convergence of sensory and pontine inputs. Survey of ECN and BPN convergence in the hemispheric regions. (**A**) Paramedian lobule, anterior section. (**B**) Paramedian lobule, posterior section. (**C**) Paraflocculus, anterior section. (**D**) Paraflocculus, posterior section. (**E**) Crus2, anterior section. (**F**) Crus2, posterior section. (i) Granule cell (GrC) classification and distribution. Red cross: *E* GrC; green cross: *EB* GrC; blue cross: *E$^+$* GrC. Scale bars, 100 µm. (ii) Density contour map of *E*, *EB* and *E$^+$* granule cells. D: dorsal; L: lateral. Red, green and blue lines in the contour map represent density of *E*, *EB,* and *E$^+$* granule cells respectively. (iii) Upper, percentage of *E*, *EB* and *E$^+$* granule cells from two *Slc17a7-IRES-Cre; TCGO* cerebella. Lower, comparison between annotators in percentage of *E*, *EB* and *E$^+$* granule cells in a selected section. (**E** and **F**) do not have comparisons across the two cerebella in (iii).

reduction (*Jorntell and Ekerot, 2006*; *Ekerot and Jorntell, 2008*). Such reassignment of function implies a marked reduction in the pattern discrimination capability of the cerebellum. In the cerebellar flocculus, *Arenz et al. (2008*, *2009)* provided indirect evidence for multimodal inputs to three granule cells. For these cells, some inputs were modulated by horizontal movement of the animal while others did not show this influence. However, whether the non-modulated inputs originated from a different mossy fiber source was not tested in these experiments, stopping short of demonstrating multimodality in granule cells. Relevant to our study, both of these electrophysiological mapping experiments failed to test the convergence of pontine and sensory pathways because they were performed in animals where the cortico-ponto-cerebellar pathway was either interrupted or suppressed.

We avoided difficulties in electrophysiologically assigning input identities by instead investigating granule cell presynaptic partners using a combination of mouse genetics, viral tracing, and anatomy. This analysis has yielded the first direct evidence that a large number of mammalian cerebellar granule

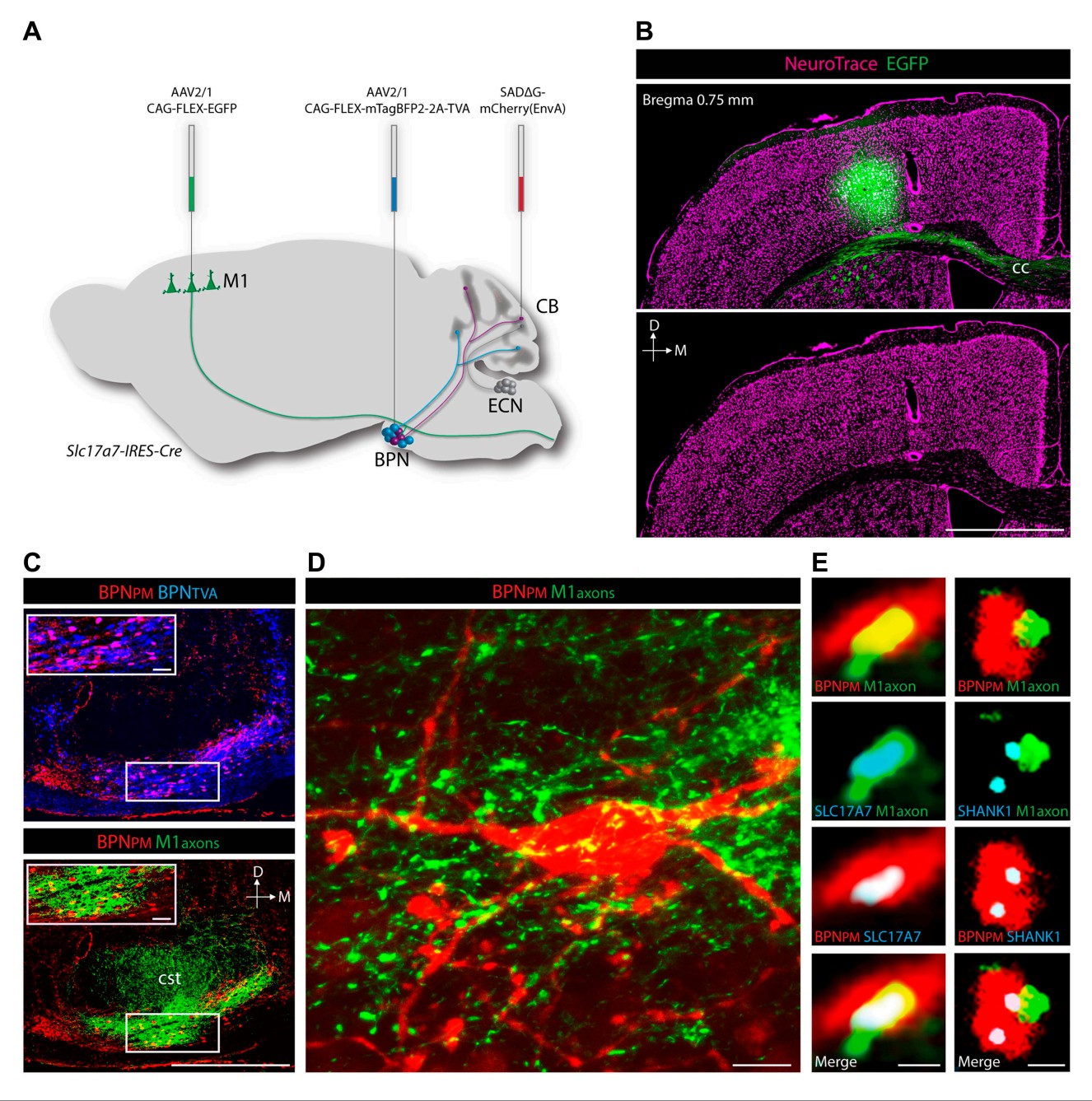

**Figure 6**. Cortical inputs to paramedian-projecting BPN neurons. Combining M1 anterograde tracing and paramedian lobule retrograde tracing.
(**A**) Scheme to anterogradely label forelimb/upper body M1 cortical axons and retrogradely label paramedian-projecting BPN neurons (BPN$_{PM}$).
Cre-dependent AAVs expressing *EGFP* and *mTagBFP2-2A-TVA* were stereotaxically injected into M1 and BPN respectively in the *Slc17a7-IRES-Cre*
mouse and SADΔG-mCherry(EnvA) rabies virus was injected into the paramedian lobule of the cerebellum. CB, cerebellum; M1, primary motor cortex.
(**B**) Location of EGFP-expressing neurons with relation to cortical cytoarchitecture. cc: corpus callosum. D: dorsal; M: medial. Scale bar, 1 mm. (**C**) Upper,
relationship of BPN$_{PM}$ and TVA-expressing BPN neurons (BPN$_{TVA}$) sensitive to rabies infection. Lower, colocalization of M1 axons and BPN$_{PM}$ neurons.
D: dorsal; M: medial. Scale bars, 500 µm; 50 µm in magnified areas. (**D**) High-magnification image of BPN$_{PM}$ neurons and the M1 axon termination field.
Scale bar, 10 µm. (**E**) Synaptic arrangement between M1 axons and BPN$_{PM}$. Left, apposition of a M1 axon expressing the presynaptic marker SLC17A7
and a dendrite of a BPN$_{PM}$ in a single confocal slice. Right, apposition of a M1 axon and a SHANK1-containing postsynaptic density of a BPN$_{PM}$ in a single
confocal slice. Scale bars, 1 µm.

cells receive inputs from both pontine and primary sensory sources. Our results also provide a map of where this variety of multimodal convergence occurs and does not occur over the expanse of the cerebellar cortex. Multimodal claw-footed granule cells have also been observed in the electrosensory lobe of the mormyrid fish, suggesting multimodality is a feature of this morphological cell type that is conserved across vertebrate species and across brain regions (*Sawtell, 2010*). The observation that our labeling strategy rarely accounted for all presynaptic partners of an individual granule cell suggests that multimodality is likely a more prevalent feature of cerebellar granule cells than indicated by the convergence of the ECN and BPN pathways reported here. By examining other precerebellar sources, systematic continuation of our mapping strategy will more fully establish the input structure and thus the associative potential of the cerebellum.

The associative capacity of the cerebellum is thought to largely depend on Purkinje cell plasticity, allowing temporal relationships among inputs to alter the strengths of postsynaptic responses (*Marr, 1969*; *Albus, 1971*; *Boyden et al., 2004*; *Gao et al., 2012*). Unimodal granule cells supply segregated input channels to Purkinje cells, allowing the molecular layer to independently adjust the weights of each modality. In addition to this independent modality control, the present finding of multimodal granule cells suggests that some parallel fiber inputs already represent information types. Prefabricating simple associations in granule cells may enable Purkinje cells to perform more complex learning operations, analogous to the enhanced learning capacity of a multilayer perceptron over a single layer perceptron (*Minsky and Papert, 1969*).

## Functional implications of pontine and sensory convergence

In order to perform skilled behaviors, motor control systems require knowledge of the environment, the position of the body, and how it moves. By signaling muscle stretch and tendon tension, proprioception is particularly well suited to provide input to the brain's sensory model of body position and kinematics (*Bosco and Poppele, 2001*; *Dietz, 2002*). Proprioceptive and other sensory afferents deliver accurate post-hoc reports of movements, but many behaviors require predicting the likely consequences of a motor plan prior to, or without, movement (*Wolpert and Miall, 1996*; *Bastian, 2006*; *Shadmehr et al., 2010*). Forward models have been proposed to use information about intended movements to predict the likely consequences of a voluntary action (*Robinson, 1975*; *Jordan and Rumelhart, 1992*; *Wolpert and Miall, 1996*). Forward models require two inputs: sensory inputs (peripheral afference) that update the state of the model and corollary discharge signals related to intended actions. To predict sensory consequences, corollary discharges are thought to mimic self-generated (reafferent) sensory information. To accomplish this mimicry, corollary discharges must be converted from motor to sensory coordinates, but where and how this occurs in the nervous system is largely unknown. Theory and perturbation studies suggest somatic forward models reside in the cerebellum (*Miall et al., 1993*, *2007*; *Miall, 1998*; *Wolpert et al., 1998*; *Blakemore et al., 2000*; *Pasalar et al., 2006*; *Ebner and Pasalar, 2008*; *Ito, 2008*; *Lesage et al., 2012*).

We show that proprioceptive and corollary discharge pathways converge on individual granule cells. Granule cells can generate action potentials in response to a single mossy fiber input (*Rancz et al., 2007*). Therefore, inputs of different modalities can potentially substitute for one another to fire a granule cell. This interchangeability suggests a plausible mechanism for converting motor corollary discharges into sensory coordinates, as outlined below. A motor command is initiated in forelimb/upper body M1 and delivered to motor output centers and the BPN by corticofugal axons. Since these BPN neurons will be driven by cortical motor commands that do not directly participate in movement generation, by definition they can be considered to carry corollary discharges. Resulting BPN output is next sent to the cerebellum to synapse with multimodal granule cells that also receive self- and externally-generated forelimb/upper body proprioceptive inputs from the ECN. Pontine corollary discharges will stimulate these granule cells, generate parallel fiber output to Purkinje cells, and may be processed similar to proprioceptive signals that synapse on the same granule cells. Although speculative, hijacking this pathway could effectively convert the corollary message from motor to sensory coordinates, and thus in the appropriate reference frame to produce proprioceptive predictions.

Corollary discharge-driven predictions will contain some degree of error, which is thought to be computed by comparing pure sensory models against forward models (*Mazzoni and Krakauer, 2006*; *Shadmehr et al., 2010*). As described above, areas containing granule cells where corollary and sensory information converge might represent components of forward models. Other areas where proprioceptive

streams are isolated from corollary/pontine information may represent what actually occurred and thus may drive pure sensory models. Common downstream targets of these identified forward and sensory models may represent loci where intended and actual actions are compared and thus where error signals are generated.

The proprioceptive pathway appears to be intersected by corollary discharges at multiple stages. Previously, corollary and sensory information has been shown to converge in the first stage of proprioceptive processing—in the precerebellar neurons of the spinal cord (*Hongo and Okada, 1967*; *Hongo et al., 1967*; *Hantman and Jessell, 2010*). The present findings indicate that one synapse further up the proprioceptive pathway—at cerebellar granule cells—this convergence occurs again. Neurons of each of these stages are distinguished by their inputs and intrinsic properties. Therefore, motor inputs at each level will produce unique transformations of the sensory streams and thus potentially embody nodes within a hierarchical motor-informed sensory processing system.

## Conclusion

Our findings confirm a key prediction made nearly a half a century ago by David Marr and James Albus about the associative faculty of the most abundant neuron type in the mammalian brain. The multimodal capacity of cerebellar granule cells calls for a systematic investigation of the possible mixtures of mossy fiber inputs throughout the cerebellum. Finally, by defining the areas of the cerebellum where corollary and sensory information potentially converge, we have uncovered a rich new system to understand the logic of corollary discharges in motor control and perhaps other brain functions.

## Materials and methods

### Mouse strains

*Slc17a7-IRES-Cre* mice were generated by the Janelia Farm-Gene Targeting and Transgenics Facility. The targeting vector contains a positive selection cassette, an *IRES-Cre* cassette and two arms homologous to exon 10 to 12 and the 3'-untranslated region of *Slc17a7* (*Figure 1—figure supplement 1*). Embryonic stem cells that correctly recombined with the targeting vector were injected into blastocysts, resulting chimeras were screened for germline transmission, and the positive selection cassette was removed by breeding F1 progeny with a codon-optimized FLP recombinase (FLPo) germline deleter strain (The Jackson Laboratory, Bar Habor, ME). *TCGO* transgenic mice were generated at Brandies University (Shima et al., under revision) by random insertion of enhancer-trap lenti-viral vectors through mouse zygote infection (*Kelsch et al., 2012*). Both mouse lines were backcrossed to the C57/B6 background. *Rosa26-loxP-Stop-loxP-lacZ* mice were obtained from The Jackson Laboratory.

### Stereotaxic viral injections

Adult mice (2–6 months old) were anesthetized with 2% isoflurane/95% oxygen mixture (VetEquip, Pleasanton, CA) and placed in a stereotaxic apparatus (David Kopf Instruments, Tujunga, CA). Following a scalp incision, small holes were drilled into the skull and the dura was exposed. Coordinates for the BPN were 4.0 mm posterior to Bregma, 0.4 mm lateral to the midline and 5.8/5.5/5.2/5.0 mm deep from dura; coordinates for the ECN were 7.2–7.4 mm posterior to Bregma, 1.3 mm lateral to the midline and 3.0/2.8 mm deep from dura; coordinates for M1 were 0.7 mm anterior to Bregma, 1.7 mm lateral to the midline and 0.75 mm deep from dura. A pulled-glass pipette (20 μm tip diameter) driven by a micromanipulator (Scientifica, Uckfield, United Kingdom) was inserted into the target area and one to four injections (50 nl per injection site) were made using a Nanoliter 2000 injector (World Precision Instruments, Sarasota, FL). For each penetration, a 2-min waiting period was imposed between sites and the pipette was slowly withdrawn 5 min after the final injection. After the surgery, the scalp was sutured, betadine was applied for antiseptic purposes, and ketoprofen (5 mg/kg) analgesic was provided subcutaneously. Mice were housed for 21–28 postoperative days in order to achieve optimal viral reporter expression and then were perfused for histology.

### Virus production

AAV2/1 CAG-FLEX-EGFP-WPRE-bGH ($1 \times 10^{13}$ particles per ml) and AAV2/1 CAG-FLEX-tdTomato-WPRE-bGH ($2 \times 10^{13}$ particles per ml) were produced by The Gene Therapy Program at the University of Pennsylvania (Philadelphia, PA). Tag-blue fluorescent protein (*mTagBFP2*; Evrogen, Moscow, Russia) and the avian virus receptor, *TVA*, were subcloned into a CAG-FLEX-2A viral vector. AAV2/1

CAG-FLEX-mTagBFP2-2A-TVA ($9 \times 10^{12}$ particles per ml) was made at the Janelia Farm-Molecular Biology Shared Resource and purified through cesium-chloride density gradients. Pseudotyped SADΔG-mCherry(EnvA) rabies virus was produced as previously described (*Wickersham et al., 2007*, *2010*) and were acquired from the Systems Neurobiology Laboratories (E. Callaway) at The Salk Institute for Biological Studies (San Diego, CA).

### Retrograde tracing by recombinant rabies viruses
Conditional expression of *TVA* in the BPN was achieved through stereotaxic injection of AAV2/1 CAG-FLEX-mTagBFP2-2A-TVA in *Slc17a7-IRES-Cre* mice. Pseudotyped SADΔG-mCherry(EnvA) rabies virus was injected into the paramedian lobule 2 weeks after the initial AAV injection. The paramedian lobule (7.0–7.4 mm posterior to Bregma, 2.3 mm lateral to the midline and 1.8/1.5 mm deep from dura) was chosen based on the convergence pattern observed from our anterograde mapping results. Animals were housed in a BSL-2 room for seven postoperative days and then perfused for histology.

### Tissue preparation and histology
Mice were deeply anesthetized with isoflurane and perfused with phosphate buffered saline (PBS) followed by 4% paraformaldehyde (PFA) in PBS. Brains were dissected and post-fixed in 4% PFA for 4 hr. Tissues were transferred to 30% sucrose in PBS for 48 hr and then embedded in Tissue-Tek OCT compound (Sakura Finetek, Torrance, CA). Brain sections (50 μm) were made using a cryostat (Leica) and mounted on glass slides in glycerol/PBS mix. Immunohistochemistry on cryostat sections was performed by sequential exposure to primary antibodies: chick anti-GFP (Abcam, Cambridge, MA), guinea pig anti-SLC17A7 (gift from the Jessell lab), rabbit anti-SHANK1 (Synaptic Systems, GmbH, Goettingen, Germany), and fluorophore-conjugated secondary antibodies (Jackson Immunoresearch, Laboratories, West Grove, PA and Invitrogen, Carlsbad, CA). NeuroTrace (Invitrogen) and 4',6-diamidino-2-phenylindole (DAPI) (Vector Labs, Burlingame, CA) were used to achieve Nissl and nuclear staining. Standard X-Gal staining protocols were used to assess β-galactosidase activity.

### Image acquisition, annotation, and data analysis
Cerebellar areas receiving ECN inputs were analyzed in two *Slc17a7-IRES-Cre; TCGO* bitransgenic animals. For each area, two positions were selected along the anterior-posterior axis. Crus2 was damaged in the second animal and was not included in this analysis. Confocal stacks were acquired by a Zeiss LSM 710 confocal microscope using 10× (0.45 N.A.) air, 20× (0.8 N.A.) air, 40× (1.3 N.A.) oil, 63× (1.4 N.A.) oil, and 100× (1.4 N.A.) oil objectives. Section boundaries were selected and stitched using the MultiTime (version 25) macro plug-in for Zen 2010 (Carl Zeiss Microimaging, Thornwood, NY). Images were acquired using a 405-nm diode laser line for mTagBFP2, DAPI, and NeuroTrace (filter setting, 391–453 nm); a 488-nm argon laser line for EGFP (488–514 nm); a 514-nm argon laser line for mCitrine (524–563 nm); a 561-nm diode-pumped solid-state laser for tdTomato (584–691 nm); a 594-nm helium-neon laser for mCherry (589–696 nm); and a 633-nm helium-neon laser for Alexa647 (638–755 nm).

Tiled image stacks were analyzed by three annotators (two for each animal) with the Zen program. Annotators identified mCitrine-positive, claw-footed structures completely embedded within a tdTomato-positive ECN rosette and manually traced back to the granule cell somata. From the soma, each of the other dendrites was manually traced to its termination and associated mossy fiber synapses were identified. Granule cells lacking another traceable dendrite were excluded from further analysis. Granule cells with at least one tdTomato ECN input and at least one other identifiable dendritic claw foot without a labeled input were marked with red crosses (*E* granule cell). Granule cells with more than two ECN inputs were labeled with blue crosses (*E⁺* granule cells). Granule cells with at least one ECN and at least one EGFP-positive rosette (BPN input) were annotated with green crosses (*EB* granule cells). For anterior vermal areas, annotators also identified mCitrine-positive dendritic arborizations embedded within an EGFP-positive BPN rosette and traced back to the granule cell soma. All other dendrites were traced from the soma and associated mossy fiber synapses were identified. In these select vermal areas, granule cells with one EGFP BPN input and at least one other identifiable dendritic claw foot without a labeled input were marked with red crosses (*B* granule cells). Granule cells with more than two BPN inputs were labeled with blue crosses (*B⁺* granule cells). Graphs showing percentage of granule cell types between animals and annotators were produced using Excel (Microsoft, Redmond, WA). X-Y coordinates of annotation markers were exported from Zen and density contour maps were made using custom Python (Enthought, Austin, TX) scripts. 3D reconstruction of granule cells and associated mossy fibers were made using Fiji-TrakEM2.

## Acknowledgements

We thank K Ritola and L Looger for assistance in the production of viruses, G Blake for efforts in annotations, and T Jessell and E Callaway for reagents. We are grateful to J Dudman, T Gonen, T Jessell, L Looger, N Spruston, K Svoboda, and C Zuker for critical comments on the manuscript.

## Additional information

### Competing interests

SBN: Reviewing editor, *eLife*. The other authors declare that no competing interests exist.

### Funding

| Funder | Grant reference number | Author |
|---|---|---|
| Howard Hughes Medical Institute | | Cheng-Chiu Huang, Adam W Hantman, Ken Sugino, Caiying Guo, Suxia Bai, Brett D Mensh |
| National Institute of Neurological Disorders and Stroke | NS075007 | Sacha B Nelson, Yasuyuki Shima |

The funders had no role in study design, data collection and interpretation, or the decision to submit the work for publication.

### Author contributions

CH, AWH, Conception and design, Acquisition of data, Analysis and interpretation of data, Drafting or revising the article; KS, Analysis and interpretation of data; YS, CG, SB, Contributed unpublished essential data or reagents; BDM, Drafting or revising the article; SBN, Drafting or revising the article, Contributed unpublished essential data or reagents

### Ethics

Animal experimentation: Mice were housed on a 12-hour light/dark cycle with ad libitum food and water access. Experimental procedures were conducted according to the National Institute of Health guidelines for animal research and approved by the Institutional Animal Care and Use Committee at Janelia Farm Research Campus. Approved animal protocol is IACUC 10-64.

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
