## [Decision Letter]

Thank you for choosing to send your work entitled “Convergence of pontine and proprioceptive streams onto multimodal cerebellar granule cells” for consideration at *eLife*. Your article has been evaluated by a Senior editor and 3 reviewers, one of whom is a member of our Board of Reviewing Editors. The Reviewing editor and the other reviewers discussed their comments before we reached this decision, and the Reviewing editor has assembled the following comments to help you prepare a revised submission.

This is an interesting paper and a potentially important contribution to our understanding of cerebellar structure and function. The data are presented clearly and precisely, and they unambiguously demonstrate that granule cells are multimodal, supporting the predictions of the Marr–Albus theory of cerebellar function. Although the main conclusions of this study are strongly supported and the results of clear importance, the paper could be improved by further discussion of three points:

* First, more description is needed of the transgenic animals that were employed for this study. For example, very little data were included or discussed regarding the specificity and efficacy of recombination of the *Slc17a7* mice. The reader also has no information regarding the TCGO transgenic mice – is the mosaic pattern of expression present in these mice stochastic, or does it represent a biologically meaningful subset of GrCs? Although these points are not of central importance for the present study, readers interested in using these mice would benefit significantly from further discussion of their properties.

* Second, the experiment in Figure 6 is of significant interest, but it is described too briefly. No quantitation is given regarding the colocalization of presynatic proteins with the marked BPN neurons, for example. Although we understand that this methodology is much less efficient than one would wish, some indication of the reproducibility of the results is warranted.

* The discussion of the meaning of an internal model, and their statements about how the convergence of corollary discharge onto sensory neurons automatically converts the motor signals to sensory coordinates are not particularly persuasive. The paper would benefit from some discussion of the implications of such an early convergence for the theory of cerebellar learning. If the sensory and corollary discharge signals are present on granule cells, then they cannot be adjusted independently by the putative learning mechanisms in the molecular layer.

In our opinion, additional discussion of these issues would enhance the paper and should be included prior to publication.

---

## [Author Response]

** First, more description is needed of the transgenic animals that were employed for this study. For example, very little data were included or discussed regarding the specificity and efficacy of recombination of the *Slc17a7* mice. The reader also has no information regarding the TCGO transgenic mice – is the mosaic pattern of expression present in these mice stochastic, or does it represent a biologically meaningful subset of GrCs? Although these points are not of central importance for the present study, readers interested in using these mice would benefit significantly from further discussion of their properties.*

In the first point, the reviewers ask for further description of the new transgenic mice used in this study. We have included a figure supplement (Figure 1—figure supplement 1) to demonstrate that our *Slc17a7-IRES-Cre* mouse line recapitulates the expression of the *Slc17a7* gene. For this figure, we crossed *Slc17a7-IRES-Cre* and *Rosa26-loxP-Stop-loxP-lacZ* mouse lines. The expression of *lacZ* reflects the expression of *Cre* in the *Slc17a7-IRES-Cre* line. These images can be compared to *Slc17a7* expression reported in the Allen Institute Anatomic Gene Expression Atlas database. We will also add brain-wide images of *Slc17a7-IRES-Cre; Rosa26-loxP-Stop-loxP-lacZ* mice to our data repository.

A discrete subset of granule cells is labeled in the *TCGO* line. The reviewers ask if the labeled neurons represent a unique subset of granule cells. The simple answer is that we do not yet know. The *TCGO* mouse line was produced by the random integration of a reporter cassette in the mouse genome (reported in a manuscript currently being revised). This raises the possibility that transgene expression is controlled by a nearby gene whose expression is restricted to this granule cell subset. Such exclusive gene expression could biologically distinguish these granule cells from other cerebellar neurons. We determined that the TCGO transgene integrated into the fifth intron of the *Tumor necrosis factor ligand superfamily member* 12 gene. However, the expression pattern of this gene (as reported in Allen Institute Anatomic Gene Expression Atlas database) is not similar to reporter expression in the TCGO line. Therefore, we have yet to identify molecular differences between labeled and unlabeled granule cells in the TCGO mouse. We are currently attempting to profile the transcriptomes of TCGO-labeled and unlabeled neurons. We have added a line to the Results section indicating the unknown mechanism of sparse granule cell labeling.

** Second, the experiment in Figure 6 is of significant interest, but it is described too briefly. No quantitation is given regarding the colocalization of presynatic proteins with the marked BPN neurons, for example. Although we understand that this methodology is much less efficient than one would wish, some indication of the reproducibility of the results is warranted*.

In the second point, the reviewers raise concerns about the reproducibility of the finding that M1 cortex provides inputs to paramedian-projecting BPN neurons. The reviewers ask for quantification of the synaptic inputs to the BPN neurons. As the reviewers suggest, such quantification is difficult. To achieve such synaptology, we used only sparse retrograde labeling and injected anterograde tracers into small focal areas of cortex. In addition, due to limited tissue penetration, the synaptic antibodies labeled only a subset of synapses. The high false negative rate of these combined methods increases the significance of positive results. Therefore we added quantification highlighting the positive results found in all animals examined in this study. We also added justification for our lack of further quantification.

** The discussion of the meaning of an internal model, and their statements about how the convergence of corollary discharge onto sensory neurons automatically converts the motor signals to sensory coordinates are not particularly persuasive. The paper would benefit from some discussion of the implications of such an early convergence for the theory of cerebellar learning. If the sensory and corollary discharge signals are present on granule cells, then they cannot be adjusted independently by the putative learning mechanisms in the molecular layer*.

In the last point, the reviewers ask to clarify our conclusions concerning internal models and motor-sensory conversion and to discuss the implications of our findings on theories of cerebellar learning. We have substantially modified the Discussion to address these issues.